# A Novel Adaptive Deskewing Algorithm for Document Images

**DOI:** 10.3390/s22207944

**Published:** 2022-10-18

**Authors:** Wuzhida Bao, Cihui Yang, Shiping Wen, Mengjie Zeng, Jianyong Guo, Jingting Zhong, Xingmiao Xu

**Affiliations:** 1School of Information Engineering, Nanchang Hangkong University, Nanchang 330063, China; 2Australian AI Institute, University of Technology Sydney, Sydney, NSW 2007, Australia

**Keywords:** skew estimation, image classification, document image, deskewing, adaptive strategy

## Abstract

Document scanning often suffers from skewing, which may seriously influence the efficiency of Optical Character Recognition (OCR). Therefore, it is necessary to correct the skewed document before document image information analysis. In this article, we propose a novel adaptive deskewing algorithm for document images, which mainly includes Skeleton Line Detection (SKLD), Piecewise Projection Profile (PPP), Morphological Clustering (MC), and the image classification method. The image type is determined firstly based on the image’s layout feature. Thus, adaptive correcting is applied to deskew the image according to its type. Our method maintains high accuracy on the Document Image Skew Estimation Contest (DISEC’2013) and PubLayNet datasets, which achieved 97.6% and 80.1% accuracy, respectively. Meanwhile, extensive experiments show the superiority of the proposed algorithm.

## 1. Introduction

Scanning is one of the most widely used ways to digitize documents. However, the skewing generated during the process of document scanning will affect the quality of document images and reduce the efficiency of OCR [1] recognition. Therefore, it is indispensable to study effective document skewing estimation and correction algorithm.

A number of document image skewing correction techniques have been reported in the literature, and most of them are mainly divided into the following categories: (1) Line detection-based methods [2,3,4,5,6]. (2) Nearest-neighbor clustering-based methods [7,8]. (3) Projection profile analysis-based methods [9,10,11]. (4) Fourier transform based methods [12,13,14,15]. (5) Axis-parallel bounding box-based methods [15,16,17,18].

Line detection-based methods calculate the skewing angle of the image through straight lines in the image. Nearest-neighbor clustering-based methods draw the histogram of the vector direction of the image to estimate the skew angle. Projection profile analysis-based methods calculate the skew angle of the image by the largest projection profile peak. Fourier transform-based methods transform each pixel from the spatial domain to the frequency domain in the image. Although this kind of method is not affected by the image content, its computational complexity is large. Axis-parallel bounding box-based methods divide the image content into blocks, wrapped with the minimum boundary rectangle (MBR). The skewing angle is estimated by calculating the angle and area of MBR.

The line detection-based method and the projection profile-based method have high accuracy for estimating skew angle on text documents. The nearest-neighbor clustering-based method and the analysis of the background of documents images-based method are suitable for correcting charts or comics. For other types of images, their correction accuracy is not high. The Fourier transform-based method is not affected by the document’s content, but its running speed is relatively slow, especially when there are too many noise and interference elements in the image. These methods are effective for document images with specific content, but they cannot be applied to any type of document while ensuring the running speed—their commonality is not high enough. Hence, in this paper, we proposed a novel adaptive deskewing algorithm for document images. The adaptive deskewing algorithm we proposed can predict the types of document images and select the appropriate algorithm for correction, which does not need any assumption concerning the document style or content. In addition to the proposed image classification algorithm, we also bring the following innovations: (1) The Image Classification (IC) method. This method can judge whether the image belongs to a text image, form image, or complex content image based on the layout feature. (2) The Skeleton Line Detection (SKLD) and Piecewise Projection Profile (PPP) method. Compared with Hough line detection, the SKLD and PPP method can effectively solve any type of text document skewing, even if there is no obvious line. (3) The Morphological Clustering (MC) method. In the face of complex content documents, the operation of clustering dramatically reduces the noise interference in the frequency domain. The illustration of the process of deskewing document images in our work is shown in Figure 1.

Experimental data include the DISEC’2013 database and PubLayNet database. In the DISEC ‘2013 dataset, our overall accuracy exceeds the best result of DISEC’ 2013 [19]. In addition, our proposed method was tested on document images taken from PubLayNet datasets. The proposed method has exhibited good performance in terms of accuracy and robustness. The correction effect for skewed images is satisfied, and the necessity of each module is proved by ablation study. Compared with the existing methods, the proposed method has stronger robustness and higher precision.

The article is divided as follows. In Section 2, we briefly review the traditional methods. Then, Section 3 discusses the details of our proposed method. Simultaneously, we present our method’s preliminary results and evaluation in Section 4 and conclude this paper in Section 5.

## 2. Related Works

Various skew detection algorithms have been proposed in recent years. Most of the algorithms, such as line detection methods and projection profile methods, can only estimate the skew angle of a specific content image, and yet for a complex content document image, their correction results are not ideal. Therefore, how to deal with document images with diverse contents is the focus of current research. In order to enhance the adaptability of the algorithm to different documents, some scholars combine two or more correction methods to achieve better results.

In 2013, the ICDAR Document Image Skew Estimation Contest [19] attracted a number of scientific research teams. The winning method of this contest uses the magnitude spectrum of a frequency Fourier transform to determine the orientation of the document image [20]. All regions of the document image are clustered using a KNN, which makes the orientation in the frequency domain easier to be detected. Considering that there are different kinds of redundant edges in document images, Cai et al. [21] proposed an improved algorithm, which can automatically crop and estimate the skew angle of the document image. Hyung IL Koo et al. [22] proposed a salient line detect-based method that highlights the edges representing the skew degree by improving the edge detection method, and then the edge line is extracted through Progressive Probabilistic Hough Transform (PPHT) [23]. Riaz Ahmad et al. [24] introduced a skew detection method in document images through the clustering of probabilistic Hough transforms, and they believed that maximum parallel lines can represent the set of the true-line. Felix Stahlberg et al. [25] proposed a skew correction method based on Hough space, which combines Hough transform and projection.

The projection of image Hough space in the vertical and horizontal directions has a peak, while the projection in other directions tends to be smooth. Therefore, the angle corresponding to the peak is the image skew angle. Based on corner features, Ahmed gari et al. [26] proposed a method by applying Hough transform to Harris interest points. In order to eliminate the interference caused by noises, Omar boudraa et al. [27] introduces a method based on morphological skeleton to eliminate redundant pixels and noise and retains the central curve of image components simultaneously. Geometric constraints are also applied to skew correction. Ju et al. [28] propose to estimate the skew angle based on the lowest point of character contour. The text area can also be used on skew estimation. Marian Wagdy et al. [29] draw a rectangular bounding box through the extreme points, and the skew angle of the rectangular bounding box is the skew angle of the document image. In order to improve the accuracy, Papandreou A et al. [30] add vertical and horizontal projections based on the minimum bounding box.

Some researchers have also proposed skew estimation and cropping methods based on deep learning. Dai et al. [31] proposed an orientation-correction detection method for scene text based on SPP-CNN (Spatial Pyramid Pooling Convolutional Neural Networks). This network effectively extracts the text features and estimates the skew angle of the text image through the extracted text features. Combining Attention Box Prediction (ABP) and Aesthetics Assessment (AA), Wang et al. [32] designed a depth network to crop the image.

Admittedly, the above correction algorithms have achieved good results in their target dataset, but they still have limitations on the types of input images. The method based on deep learning is to pay more attention to the text content rather than its skewing degree. In order to increase the recognition efficiency, they may roughly estimate and correct its skewing angle. Obviously, the accuracy of correction cannot meet the industrial demand (<0.1°). Our method is superior in the breadth of image types and the accuracy of correction.

## 3. Method

### 3.1. Algorithm Overview

Initially, the type of document image is determined through the image classification method. After that, the algorithm utilizes the most appropriate strategy to estimate the skew angle according to the type of the document image. Finally, the skew image is corrected according to the estimated angle. The detailed steps are shown in Figure 1.

**The correction of the text image.** For text images, we combine skeleton line detection (SKLD) with the Piecewise Projection Profile (PPP) method for correction. After obtaining the writing direction of the text, we used SKLD for correction in the first step, and then the PPP method was used to correct the image in the second step. The image skew angle is calculated by adding the estimated skew angle of the aforementioned steps.

**The correction of the form image.** Based on the table lines, the skew angle of the form image is estimated by the Hough line detection method. Firstly, the table line is detected, and the outlier is filtered in the line set. Then, the average skew angle of the filtered line set is calculated, and this angle is considered as the skew angle of the table document image.

**The correction of the complex content image.** Compared with the other two types of images, it is more challenging to analyze the layout information in complex content images. Therefore, the Morphological Clustering (MC) method combined with Fourier transform is used to estimate the skew angle of the document image. The adjacent text regions are clustered by MC, and their outermost outlines are extracted. Then, frequency domain maps are obtained through Fourier transform the outline images. The skew angle can be estimated according to the high-frequency characteristics of the frequency domain image.

In the end, the skew image is corrected according to the calculated angle.

### 3.2. Image Classification

According to the layout features of document images, images can be divided into text images, form images, and complex content images. The layout features of most text images are neatly arranged, and the form image contains a lot of clear table lines. Complex content images have complex layouts and varying line spacing. Therefore, the categories of the image can be distinguished by arrangement rules and detected lines. The flow of image classification is shown in Figure 2.

Initially, the foreground and background of the content are distinguished by using adaptive binarization. Then, the binary image is morphologically processed with a *M × N* structure element to fill the blank inside the text and connect the adjacent text areas. We find the contour of all elements in the image and calculate the aspect ratio and size of each contour. The contour with aspect ratio *w/h* > 2 is marked as horizontal contour *C_v_*, and other contours are marked as vertical contour *C_h_*, where *w* and *h* are the width and height. If *C_h_/C_v_* is greater than *t_Max_* or less than *t_Min_* (*t_Max_* and *t_Min_* are Maximum aspect ratio and Minimum aspect ratio), the image is marked as a text image (*T*). Otherwise, it will be marked as a non-text image (*NT*), as shown in Equation (1). Because most of the text lines are rectangular regions after morphological processing, and in order to eliminate the effect of some noise contours on the algorithm, we empirically set *t_Max_* = 3 and *t_Min_* = 1/3, which are the most suitable for filtering text images.
(1)$$\left\{\begin{array}{l} T\;r\notin \left(t_{Min},t_{Max}\right)\\ NT\;r\in [t_{Min},t_{Max}] \end{array}\right.,r=C_{h}/C_{v}$$

The non-text image is further subdivided according to the number of the detected lines. When the number of lines exceeds the threshold, we mark it as a form image. Otherwise, it is marked as a complex content image. Firstly, we adopt the Canny operator to traverse the binary image. Then, PPHT is used to detect the line of the image. In this paper, to ensure the quality of the line, the *minVotes*, *minLineLength,* and *maxLineGap* used in PPHT are fixed as 30, 50, and 5 respectively. When the number of lines *l* detected in the image is more than *L_Max_*, and the variance of line slope *var*(*k*) is less than *Var_Min_*, the image is marked as a form image (*FI*), as shown in Equation (2). Otherwise, it will be marked as the complex content image (*CCI*). After extensive experiments, we set *L_Max_* and *Var_Min_* to 6 and 10.
(2)$$NT=\left\{\begin{array}{l} FI\;if\;l\geq L_{Max}\;and\;var(k)<Var_{Min}\\ CCI\;if\;l<L_{Max}\;or\;var(k)\geq Var_{Min} \end{array}\right.$$

The result of image classification is shown in Figure 3. In general, we have the following definitions for these three types of images. Text image: An image containing a sufficient number of text lines that meet the requirements. Form image: In addition to text images, images containing a sufficient number of table lines or straight line segments that meet the requirements. Complex content image: Images other than text images and form images.

### 3.3. Text Image Correction

**Text writing direction judgment.** For a text image, the direction of text writing determines the direction of the text line, and the skew angle of the image can be determined by detecting the skew angle of the text line. Most English documents are written horizontally. However, there are also some documents written vertically, such as poetry, invitations, etc. For projection or morphological transformation, the effect of processing along the transcendental text line direction will be more accurate. Therefore, we divide the text writing direction into horizontal and vertical. Particularly, the algorithm proposed in this paper limits the tilt angle of the input image to (−45°, 45°). This is because the image beyond this angle has been considered to face other directions instead of being skewed.

Most of the pixels are continuous along the text line, while the pixels vertical to the text line are sparse. Therefore, we can judge the writing direction of the text by the slope of the text line. Since the text has only two writing directions, there is no need for a highly accurate line detection method, so we used LSD line detection [33]. Compared with the Hough transform, LSD line detection is faster.

The LSD algorithm is used to detect the line, and the direction of the line (DOL) is judged by its slope. When the slope of the line is greater than 1, we add this line to vertical line set or else add it to horizontal line set. By comparing the number of lines in horizontal and vertical sets, we can judge the writing direction of the text. We denote the quantity corresponding to horizontal lines and vertical lines by *L_h_* and *L_v,_*, respectively. When *L_h_ > L_v_*, the text is writing horizontally; otherwise, it is writing vertically.

**Skeleton Line Detection.** Because of the lack of lines in text images, the correction effect of the method based on line detection is ineffective. By observing a large number of samples, we found that in nearly all cases, the lines of text are parallel to each other, and the lines of text are parallel to the boundaries of the image. Technically, it is possible to estimate the skew angle of an image from the skew angle of a text line. In order to detect the inclination of text lines, we utilize the image thinning algorithm of Zhang-Suen to extract the text skeleton and then predict the inclination of text lines by detecting the text skeleton lines. Even if there is no line in the text image, the method can still predict the image skew angle through the text line, which significantly enhances the method’s robustness. The operation steps of the skeleton line detection method are as follows.

For a text image, we should determine the writing direction of the text firstly. Then, according to the writing direction of the text, we determine the size of the *M* and *N* for expanding the binary image, which are defined as follows: When the text writing direction is vertical, we determine *M > N*; Otherwise, we determine *M < N*. Figure 4b illustrates the result of the binary image after expansion. For getting effective contours, we retain text contours whose aspect ratio is greater than *x* when we detect text contours in the binary image, where *x* is the minimum value of the aspect ratio of the profile. After that, we draw the contours’ minimum bounding rectangles on the binary image of the same size. The result is shown in Figure 4c. Finally, the image thinning algorithm is used to extract the skeleton of the minimum bounding rectangle of text lines, and the result is shown in Figure 4d. The skew angle of the image can be calculated through the slope of lines on the skeleton.

**Piecewise Projection Profile.** Projection profile is a classical image correction method. However, continuous projection will take much time, and the position of the text area corresponding to the peak value is constantly changing, which is not suitable for the evaluation standard of profile comparison. Therefore, we propose a Piecewise Projection Profile (PPP) method based on valley value to optimize these defects.

Traditionally, researchers would utilize peak as the evaluation standard of the inclination angle. However, after experiments, we found that the traditional peak projection method is unstable when correcting the image with excessively dense content. The peak value at a certain angle may be larger than the peak corresponding to the real angle when the content structure is excessively dense. In order to solve this problem, we utilize the number of black pixels in the red area as the benchmark to determine the inclination angle, as shown in Figure 5. We call this benchmark valley value. Within a certain rotation range, the rotation angle corresponding to the minimum valley value is the tilt angle of the image.

Compared with the traditional projection profile method, we divide the projection process into two steps. For the first segment projection, we use a large angle *L*_1_ to rotate and project the image to obtain the image skew angle α. For the second segment projection, we take the range $\left[\alpha -L_{1},a+L_{1}\right]$ as a new rotation range and define a smaller angle *L*_2_ as the rotation angle. In our experiments, we empirically set *L*_2_
*= L*_1_/10. The skew angle obtained by the second step is the skew angle of the image. This method is referred to as PPP, and the specific steps of the proposed method are as follows:Scale the image equally, as shown in Equation (3). Then, distinguish the foreground and background of the image by using adaptive binarization algorithm.
(3)$$ratio=\left\{\begin{array}{c} R/w_{ori}\;w_{ori}>h_{ori}\\ R/h_{ori}\;w_{ori}\leq h_{ori} \end{array}\right.$$
where *w**_ori_* and *h_ori_* are the width and height of the original image, and ratio is the scaling ratio; *R* is set to *1800* in this article.

2.For the first segment projection, let $[\theta_{start},\theta_{end}]$ be a rotation range, and denote by *L_1_* the rotation angle’s interval. In this paper, we set *L_1_* = 0.1°, *θ_start_ =* −0.5° and *θ_end_ =* 0.5°. The projection direction is selected according to the text writing direction. If the text writing direction is horizontal, the document image is projected horizontally to obtain the horizontal projection profile. Otherwise, it is projected vertically to get a vertical projection profile.3.Calculate the valley value of the projection profile and find the angle *θ* corresponding to the minimum valley value. *Θ* is the skew angle of the image when the accuracy is *L*_1_. For example, the valley value (*Val*) of the horizontal projection is calculated as shown in Equation (4).(4)$$Val=\sum_{i=5}^{9}\sum_{j=0}^{h}P(i,j)$$
where *P*(*I*,*j*) represents the pixel with coordinate (*i,j*) on the projection profile, *h* is the height of the projected profile. As shown in the red box of Figure 5, the value range of *i* in this paper is [5,9].

4.If rotation angle *θ* is more than one when *Val* is the smallest, the starting angle of the new range $\theta_{start}^{1}$ is the smallest angle which is denoted by *θ_min_* in rotation angles, and the end angle of the new range $\theta_{end}^{1}$ is the largest angle which denoted by *θ_max_* in rotation angles. The rotation range of the second segment projection is $[\theta_{start}^{1},\theta_{end}^{1}]$, as shown in Equation (5).


(5)
$$\left\{\begin{array}{c} \theta_{start}^{1}=\theta_{min}\\ \theta_{end}^{1}=\theta_{max} \end{array}\right.if\;num(\theta )>1$$


If there is only one rotation angle, the $\left[\theta -L_{1},\theta +L_{1}\right]$ is the rotation range of the second step. We defined the rotation range of the second step as $[\theta_{start}^{2},\theta_{end}^{2}]$. The calculation is shown in Equation (6).
(6)$$\left\{\begin{array}{c} \theta_{start}^{2}=\theta -L_{1}\\ \theta_{end}^{2}=\theta +L_{1} \end{array}\right.if\;num(\theta )=1$$
where *num(**θ)* is the number of angles *θ*.

5.Set the rotation angle *L*_2_ = *L*_1_/10, and repeat the operation in step (3) according to the rotation range obtained in step (4). The angle *θ* finally predicted is the skew angle of the image. If there are multiple *θ*, we take the mean value $\bar{\theta }$ of *θ* as the final skew angle.

Compared with the traditional projection profile method, the piecewise projection profile can save nearly 80% of the calculation time. For instance, within the image skew range of −0.5° to 0.5°, the traditional projection profile method needs to rotate the image at least 100 times to achieve a 0.1° calculation accuracy, whereas using PPP, setting *L*_1_ as 0.1° and *L*_2_ as 0.01°, only 20 times of rotation is needed to achieve the same effect.

The whole procedure of the proposed piecewise projection method is summarized in Algorithm 1.
**Algorithm 1:** Piecewise projectionInput: The document image that has been pre-corrected by line detection correction.Start    resize the image    from *θ_start_* to *θ_end_* stride: *L*_1_       Project the image to the prior text writing direction.       Calculate the valley value of the project profile in each projection, and find the smallest one.                               Val=∑i=59∑j=0hP(i,j)  horizontal∑j=59∑i=0wP(i,j)  vertical
       Calculate the new projection angle interval [θstart*, θend*] based on the minimum valley value.    From θstart* to θend* stride: *L*_2_       Project the image to the prior text writing direction.       Calculate the valley value *(Var)* of the project profile in each projection and find the smallest one.    Estimate the skew angle *θ* based on the minimum valley value.    Deskew image.End

### 3.4. Form Image Correction

**Line detection and Outlier elimination.** After obtaining the line set, we first calculate the skew angle of the line. We are carrying the slope *k* of the line into the inverse tangent function formula *Arctan()* to calculate the corresponding skew angle, as shown in Equations (7) and (8).
(7)$$k=\frac{y2-y1}{x2-x1}$$
(8)$$\theta =Arctan(k)*\frac{180}{\pi }$$
where *θ* denotes the inclination angle, *k* is the slope of the line, and *x*_1_, *y*_1_, *x_2_*, and *y*_2_ are the coordinates that represented two endpoints of the line.

Besides, in order to process text in different directions, we have to convert the skew angle of the line to the same direction. As shown in Equation (9):(9)$$\theta =\left\{\begin{array}{ll} \theta +90{}^{\circ} & if{}^{\circ}\theta {}^{\circ}<-45{}^{\circ}\\ \theta -90{}^{\circ} & if+45{}^{\circ}<\theta \\ \theta & if-45{}^{\circ}\leq \theta \leq +45{}^{\circ} \end{array}\right.$$

After normalizing angles to the same direction, we need to eliminate angles that are outside the specified interval $\left[\bar{\theta }-a,\bar{\theta }+a\right]$. $\bar{\theta }$ is the mean angle, which can be calculated as Equation (10):(10)$$\bar{\theta }=\frac{1}{n}\sum_{i=1}^{n}\theta_{i}$$
where *n* is the number of lines, and *a* is the threshold of the interval. In this paper, *a* is set to 0.5°. The lines whose skew angle is within the specified interval are retained, and the mean angle of these lines is taken as the image skew angle.

### 3.5. Complex Content Image Correction

**Morphological Clustering** Due to the unique layout structure and the lowest classification priority of the complex content image, none of the above methods can achieve a good correction effect. Therefore, we collect the high-frequency features of the image in the frequency domain space to estimate its skew angle. However, irregular layout and noise in complex content images will seriously affect frequency-domain feature extraction. In order to reduce the interference of these factors, we need to perform morphological clustering of elements in the image before the Fourier transform.

Firstly, the binary image is morphologically processed with a *M × N* structure element. The adjacent elements are connected into a whole connected area. Then, we collect the contours of connected areas in the image and then filter the contours with an area smaller than *φ*:(11)$$contour=\left\{\begin{array}{c} ac\\ ec \end{array}\right.\begin{array}{c} if\;Area(contour)\geq \varphi \\ if\;Area(contour)<\varphi \end{array}$$
where *φ* is a parameter for distinguishing large area from small area, *ac* is the contour whose area is greater than *φ,* and *ec* is the contour whose area is less than *φ*. *Area(),* which is only associated with contour, is for calculating the area of this contour. In this article, *φ* is empirically set as 100.

After that, we create a new blank image *I_c_* with the same size as the original image and draw all contours marked as *ac* into *I_c_*. Next, we obtain the frequency-domain image through the Fourier transform of *I_c_* and then detect lines in the frequency-domain image. Finally, we estimate the skew angle of the image from the average skew angle of these lines.

In application, we find that the direction of the frequency domain is evident through the dilation and contour extraction, as shown in Figure 6.

## 4. Experiments

For the performance evaluation, we conduct extensive experiments on two well-known benchmarks datasets which contain many different types of inclined document images as shown in Figure 7. Results on DISEC 2013 and PubLayNet images show that the average time taken by our method to deskew a form image will approximately cost 0.21 s, to deskew a text image will approximately cost 0.83 s, and to deskew a complex content image will approximately cost 1.51 s. The time consumption of the adaptive image classification algorithm is about 0.34 s for each image. In addition, we define three evaluation metrics as indicators to evaluate the algorithm. See Section 4.2 for details.

In Section 4.3, we compared the algorithm proposed in the text with four classical skew correction algorithms using the PubLayNet dataset to prove its accuracy. At the same time, we used 0.1° as the threshold of angle estimation and compared it with the top three algorithms of DISEC 2013 and the algorithms using DISEC 2013 datasets in recent years. The results show that our algorithm has reached the leading level.

### 4.1. Datasets

For testing, we selected a large number of document images from the following datasets, including diverse types of image data and some special cases, such as documents in vertical or horizontal writing direction, pictures, charts, newspapers, and document images in many different languages.

The first test set contains 200 images extracted from the DISEC 2013 dataset [19]. These images are nominated from the benchmark test set and used by contestants to test their algorithm performance. The dataset includes images with different writing directions, different languages, and different contents. These images are rotated at ten angles randomly in the range of −15° to 15°.

The second test set consists of some images in the PubLayNet dataset [34]. The dataset includes vertical and horizontal charts, text, even color images, formulas, etc. We randomly rotated these images between −20° and 20° to create 2114 skewed document images. Figure 7 is an example of a part of the test image.

### 4.2. Evaluation Criteria

In this section, we use the following metric to evaluate the efficiency of the algorithm.

the Average Error Deviation (AED):(12)$$AED=\frac{\sum_{j=1}^{N}E\left(j\right)}{N};E(j)=abs[Pre(j)-GT(j)]$$
where *j* is the document image, *E(j)* is the distance between the ground-truth, *N* is the total number of images in the datasets, *Pre(j)* is the angle of the algorithm correction result, and *GT* is the ground truth angle.

The Average Error Deviation of the Top 80% (TOP80):(13)$$TOP80=\frac{\sum_{j=1}^{N*80\%}sE\left(j\right)}{N*80\%}$$
where *j* is the document image. The results of *E(j)* are arranged in ascending order, and the first 80% of the data are added to *sE(j)*.

The percentage of Correct Estimations (CE):(14)$$CE=\frac{\sum_{j=1}^{N}K\left(j\right)}{N};K(j)=\{\begin{array}{l} 1\;if\;E(j)\leq 0.1\\ 0\;otherwise \end{array}$$

The threshold is set to 0.1°. That is because the skew angle of greater than 0.1° can be easily observed by a human.

Moreover, for each method, we calculate its ranking according to the above metric. Then, the cumulative ranking values of the three criteria are sorted, and the final ranking is calculated.
(15)$$S=\sum_{j=1}^{3}R(j)$$

Specifically, *R*(*j*) is the ranking of the method under each evaluation standard. We define AED, TOP80, and CE as evaluation standards 1, 2, and 3, respectively. *S* is the total ranking after accumulating the ranking. The smaller the value of *S*, the stronger the performance of the corresponding algorithm.

### 4.3. Experimental Result

The accuracy of image classification algorithm is shown in Table 1. Due to all kinds of images being tested together, when one image is misjudged, the accuracy of the two types of images will be affected. We extracted the same number of text images, form images, and complex content images from the DISEC’2013 dataset, and the image classification algorithm is tested with these images. Experimental results show that the algorithm can distinguish different types of images with high accuracy. However, we do not think that some of the misjudgment images are caused by the defects of the classification algorithm. Instead, the classification algorithm is based on a given threshold to determine the type of the image. Therefore, it is possible that for some complex content images, the classification algorithm will judge them as text images when the number of text lines in the image is enough to calculate the skew angle. Yet, this judgment does not affect the operation of the subsequent correction algorithm; on the contrary, it can save more time, because the correction algorithm for the text image is faster than the algorithm for the complex content image.

From Table 2, we can observe that the CE achieved the highest results when the document image adopts the appropriate correction strategy. It is worth noting that the correction precision of these three strategies for text images is significantly higher than complex content images because text document images have the highest classification priority, and most images containing text can be classified into this category. Only images that are difficult to be processed by other strategies will be considered as complex content images.

In order to prove the validity of valley value, we use peak value and valley value as evaluation criteria of piecewise projection for the same group of text images. The results are shown in Table 3. As shown in the table, the correction accuracy of piecewise projection algorithm based on valley value is better than that based on peak value. This is because valley values are designed based on assessing the degree of alignment of text projections. For a text document with a horizontal writing direction, when the document is not skewed, the projection profile overlap of the text area in the horizontal direction is the highest, and the valley value is the minimum. In other words, when the skewing of the image is 0°, its valley value is always the smallest. (Image skew range is (−45°, 45°)).

To verify the performance of the algorithm, we compared our proposed algorithm with some traditional algorithms, including k-nearest neighbor clustering, Fourier transform, Hough transform, and projection profile analysis. The test set consists of 2114 skew document images selected from the PubLayNet dataset. Then, the proposed algorithm and the traditional algorithm were used to estimate the skew angle. As shown in Table 4, our algorithm almost perfectly estimates the skew angle of the image and far surpasses other algorithms according to the evaluation criteria proposed in Section 4.2. Table 5 shows the overall ranking of participating algorithms.

Furthermore, we also compared our method with the top three algorithms of DISEC 2013 and other algorithms based on the DISEC’2013 data set in recent years. The strategies adopted by these algorithms are very representative and demonstrate a state-of-the-art position at the time of publication. Table 6 shows the evaluation results of all participating methods, where we can see that our proposed method is less effective than other methods for AED. However, in terms of TOP80 and CE, our algorithm is optimal. Some algorithms lack some evaluation criteria, so we replace the ranking of missing evaluation criteria with the ranking of AED. Finally, we sorted S from small to large and got the overall ranking of the participating methods, as shown in Table 7.

Due to the lack of data, some methods use the AED ranking as the ranking of the missing part. It can be seen from the above ranking that our algorithm has the highest accuracy. In Figure 8, we show the comparison between the predicted value and the actual value of the skew angle of different content images.

In addition, ablation experiments were performed to demonstrate the necessity of each module. To facilitate combination, we divided the method into the following modules: Line Detection, Skeleton Extraction, Piecewise Projection, and Morphological Fourier Transform. Under the same condition, we use image classification as a baseline module and design different modules as follows.

M1: Baseline with Line Detection.M2: Baseline with Line Detection and Skeleton Extraction.M3: Baseline with Line Detection, Skeleton Extraction, and Piecewise Projection.M4: Baseline with Line Detection, Skeleton Extraction, and Morphological Fourier Transform.Ours: Refers to the method we proposed.

The experiment was carried out on the PubLayNet data set, and the results of different modules are shown in Table 8. It can be seen that our algorithm has adopted the most efficient combination and has been successfully applied to document images with different contents.

## 5. Conclusions

In this paper, we presented a novel adaptive deskewing algorithm for document images, which determines the type of document image firstly and then selects the appropriate strategy to correct it according to the type of image. This algorithm also determines the text direction of the document image and passes it as an important parameter to the subsequent strategies to select a direction which is more suitable for projection. In general, this paper brings the following innovations: on the one hand, we have proposed an image classification algorithm based on layout features. On the other hand, we have proposed SKLD, PPP, and MC methods for correcting different types of document images.

The experimental results and ablation study on the DISEC’2013 and PubLayNet show that our algorithm has high accuracy and robustness. The algorithm has good results for document images with different contents in two data sets. In addition, our proposed algorithm has some insufficiencies: the strategy is too complex and can only estimate the skew within, beyond which the image may be inverted.

In the future, we intend to improve the algorithm against these shortcomings, simplify the original strategy, and reintegrate the duplicate modules. Additionally, Gilles Simon et al. [35] proposed a generic document image dewarping method by probabilistic discretization of vanishing points. In [36], Zhai et al. gave a vanishing points detecting method through global image context in a Non-Manhattan world, and Li et al. designed a patch-based CNN for Document rectification and illumination correction [37]. We intend to make a profound study of our subject based on these methods in the future.

## Figures and Tables

**Figure 1 sensors-22-07944-f001:**
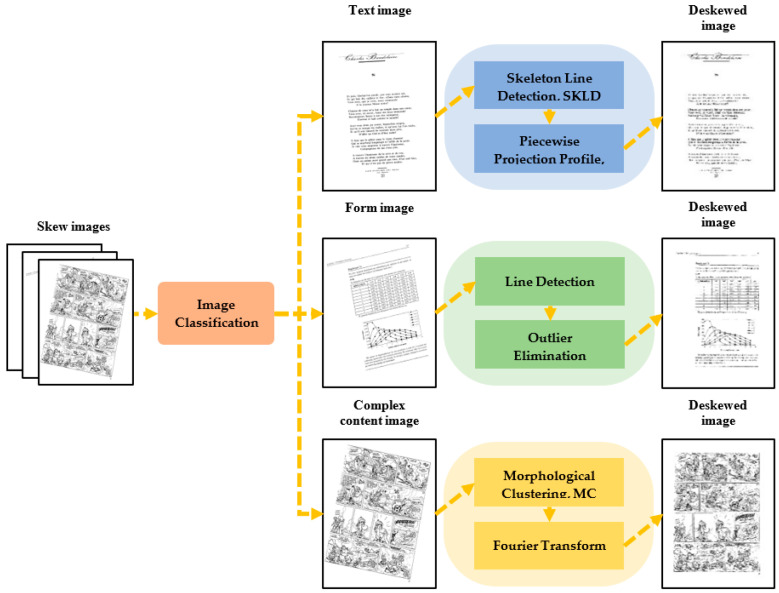
Illustration of the process of deskewing document image in our work.

**Figure 2 sensors-22-07944-f002:**
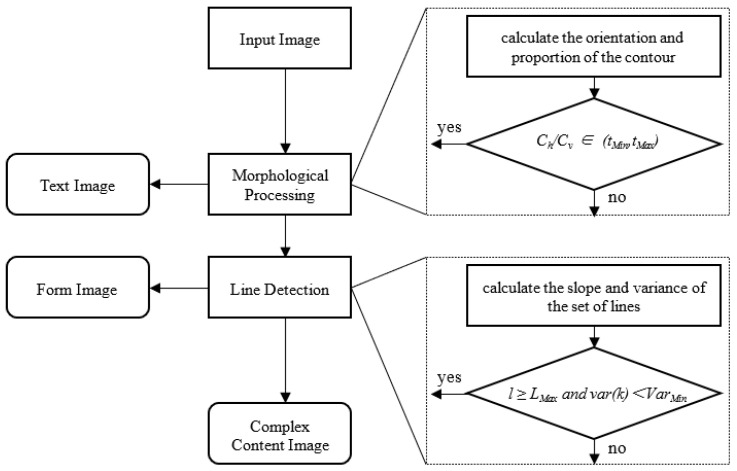
Working mechanism of Image classification.

**Figure 3 sensors-22-07944-f003:**
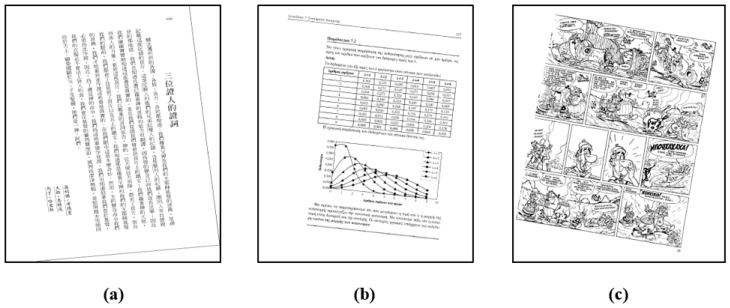
Results of image classification. (**a**) Text image (**b**) Form image (**c**) Complex content image.

**Figure 4 sensors-22-07944-f004:**
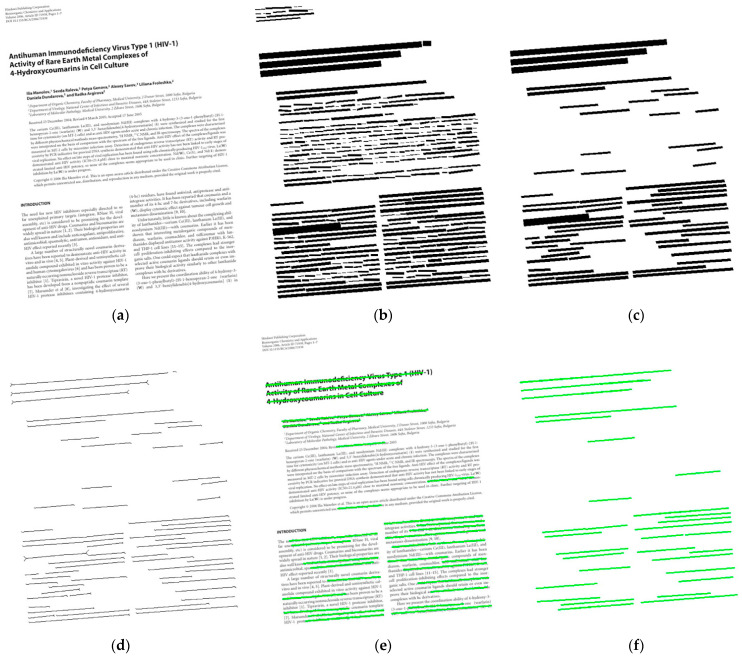
Skeleton line detection process. (**a**) Original image. (**b**) Minimum bounding rectangle image. (**c**) Filtered minimum boundary rectangular image. (**d**) Text line skeleton. (**e**) Line detection results based on text skeleton. (**f**) Text skeleton lines.

**Figure 5 sensors-22-07944-f005:**
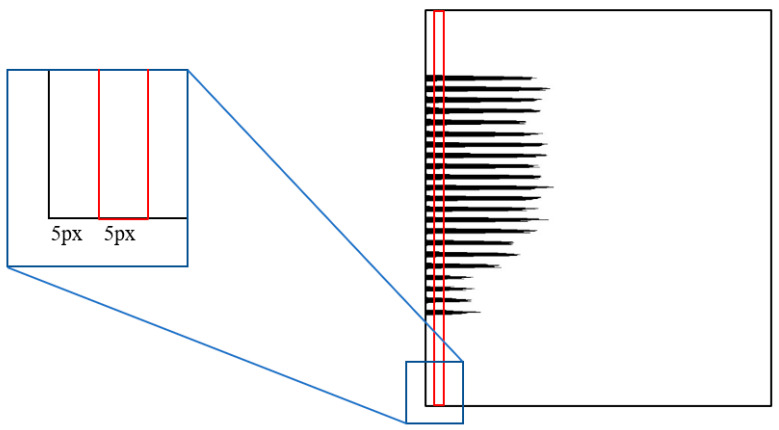
Valley value calculation area. Red box: Valley value calculation area (A rectangular area with a width of five pixels) Black pixel: The foreground pixel (Obtained by projecting the text area to the horizontal direction). The vertical axis of the projection profile image corresponds to the vertical axis of the original image, and its horizontal axis is the accumulated value of the foreground pixel on the vertical axis.

**Figure 6 sensors-22-07944-f006:**
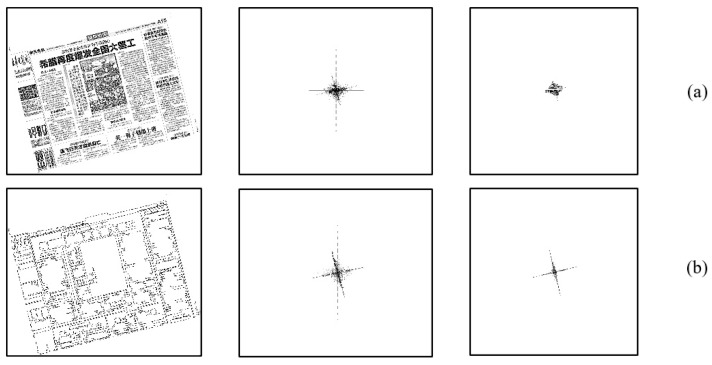
The comparison of frequency domain diagram. (**a**) Without morphological treatment. (**b**) After morphological processing. The upper part on the left side is the original image, and the lower part on the left side is the morphological clustering image. In the middle position are the frequency domain images. The images on the right side are binary images in frequency domain space.

**Figure 7 sensors-22-07944-f007:**
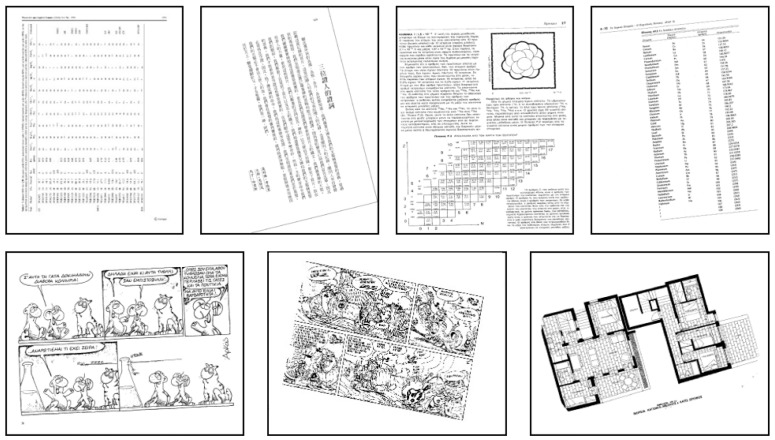
Test images from data set DISEC 2013 and PubLayNet.

**Figure 8 sensors-22-07944-f008:**
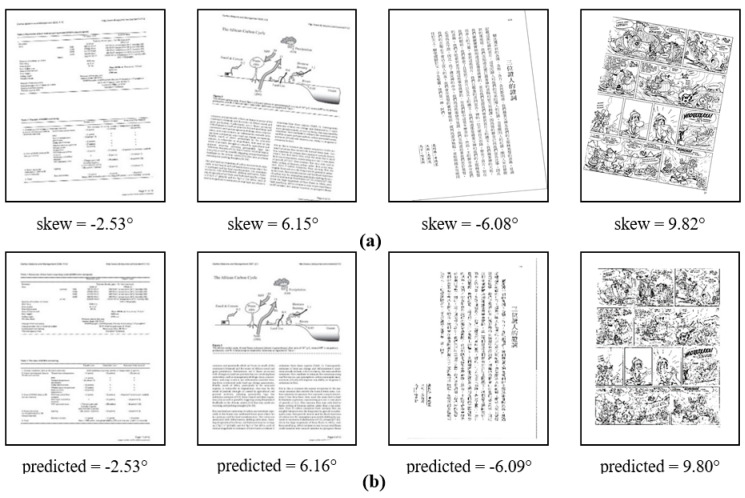
Deskewed images obtained from DISEC’2013 dataset and PubLayNet dataset. Where (**a**) is the original image and (**b**) is the image corrected by the algorithm proposed in this paper.

**Table 1 sensors-22-07944-t001:** Precision of image classification algorithm.

Skewed Image	Precision (%)
Text images	96.97
form images	100
Complex content images	96.97

**Table 2 sensors-22-07944-t002:** Matching degree detection of three strategies.

Skewed Image	Method	AED (°)	TOP80 (°)	CE (%)
Text images	Text image correction	**0.065**	**0.042**	**83.0**
	Form image correction	0.137	0.077	55.3
	Complex content correction	0.570	0.322	39.8
Form images	Text image correction	0.631	0.107	48.2
	Form image correction	**0.084**	**0.066**	**77.0**
	Complex content correction	0.949	0.781	31.2
Complex content images	Text image correction	1.414	1.050	16.1
	Form image correction	1.127	0.815	27.4
	Complex content correction	**0.055**	**0.045**	**71.7**

**Table 3 sensors-22-07944-t003:** Comparative experimental results of peak and valley values.

Method	AED (°)	TOP80 (°)	CE (%)
Peak value	0.128	0.081	60.4
Valley value	**0.065**	**0.042**	**83.0**

**Table 4 sensors-22-07944-t004:** Comparison results for the methods Fourier transform (HT), Hough transform (HT), Projection profile analysis (PP), Nearest-neighbor clustering (NNC), and our method for using the PubLayNet dataset.

Method	AED (°)	TOP80 (°)	CE (%)
FT	0.109	0.062	64.2
HT	0.095	0.053	72.2
PP	0.072	0.046	78.8
NNC	0.079	0.054	73.1
Our method	**0.025**	**0.014**	**97.6**

**Table 5 sensors-22-07944-t005:** Overall ranking on PubLayNet dataset.

Method	AED	TOP80	CE	S	Overall Rank
FT	5	5	5	15	5
GT	4	3	4	11	4
PP	2	2	2	6	2
NNC	3	4	3	10	3
Our method	1	1	1	3	1

**Table 6 sensors-22-07944-t006:** Comparison results for Chengtao Cai’s method (CCM), Omar boudraa’s method (OBM), Felix Stahlberg’s method (FSM), Riaz Ahmad’s method (RAM), and other algorithms based on the DISEC’2013 dataset with our method for using the DISEC’2013 dataset.

Method	AED (°)	TOP80 (°)	CE (%)
CCM [21]	0.083	/	68.00
OBM [27]	0.078	0.051	/
FSM [25]	0.115	0.049	73.74
RAM [24]	0.370	0.079	55.41
LRDE-EPITA-a ^1^	**0.072**	0.046	77.48
Ajou-SNU ^1^	0.085	0.051	71.23
LRDE-EPITA-b ^1^	0.097	0.053	68.32
Our method	0.077	**0.045**	**80.10**

^1^ Top three algorithms of DISEC 2013.

**Table 7 sensors-22-07944-t007:** Overall ranking on DIESC’2013 dataset.

Method	AED	TOP80	CE	S	Overall Rank
CCM [21]	4	(4)	6	14	6
OBM [27]	3	4	(3)	10	3
FSM [25]	7	3	3	13	4
RAM [24]	8	8	8	24	8
LRDE-EPITA-a ^1^	1	2	2	5	2
Ajou-SNU ^1^	5	4	4	13	4
LRDE-EPITA-b ^1^	6	5	5	16	7
Our method	2	1	1	4	1

^1^ Top three algorithms of DISEC 2013.

**Table 8 sensors-22-07944-t008:** Evaluation results of method combination.

Modules	*M* _1_	*M* _2_	*M* _3_	*M* _4_	*Ours*
Line Detection	√	√	√	√	√
Skeleton Extraction	×	√	√	√	√
Piecewise Projection	×	×	√	×	√
Morphological Fourie Transform	×	×	×	√	√
CE	72.2%	80.1%	88.4%	83.9%	**97.6%**

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
