# Peer review of "A Novel Adaptive Deskewing Algorithm for Document Images"

_sensors, 2022, doi:10.3390/s22207944_

Round 1

Reviewer 1 Report

In this paper, a new adaptive document image skew algorithm is proposed to correct skewed files and reduce the quality loss during document transmission. There are problems that need to be solved.

1. Abstracts must be better expressed on the basis of comparative numerical analysis.

2. The formula ranking in the paper must be recalibrated (equations 14, 15).

3. The reference format should be unified.

4. For equations 1 and 2, the selection of threshold should be explained.

5. Figure 4 should be reformulated to better express the sequence.

6. Check English grammar and spelling errors, and proofread the correspondence between the text and the diagram (Table 2).

Reviewer 2 Report

The method has a number of parameters - some analysis or guidance on setting these would be helpful.

Reviewer 3 Report

The article does produce excellent results, but there are many details that need to be revised:

1.       (Line 10) The emergence of 5G technology is not strongly related to document skew correction. Please clarify this point.

2.       (Line 50) The authors mention here that methods based on nearest neighbor clustering, image background analysis, and Fourier transform are not affected by the content of the document. But then they say that they are effective for specific document images, which is contradictory.

3.       (Line 56) The authors proposed algorithm can provide predictions about document style and content for subsequent steps, rather than requiring no assumptions.

4.       (Line 58) Document pre-classification and image classification refer to the same approach in this article. Please recheck the manuscript and unify the terms.

5.       (Line 120) Lack of a summary of the methods mentioned in related works, as well as a comparison with own work.

6.       (Line 156) When w/h>2, the contour is horizontal, not vertical.

7.       (Line 159) If an image contains both forms and text, it will be considered as a text image as long as the condition Eq1 is met. Whether such images are judged as text or forms will have a more significant impact on the results.

8.       (Line 189) If an image is written horizontally, but the skew is more than 45 degrees, it will be mistakenly considered to be written vertically. Here the authors implicitly assume that the skew angle is less than 45 degrees (the skew angle has never exceeded 45 degrees in the subsequent experimental design), the assumption should be specified here.

9.       (Line 224) The authors should explain the meaning of black pixels and vertical axis.

10.   (Line 327) Authors should indicate the time consumed by the adaptive image type judgment framework.

11.   (Line 351) The author should supplement the accuracy of the adaptive framework for image classification and make an analysis.

12.   (Line 382) The authors should analyze the reasons for the improved accuracy of the valley values algorithm.

13.   (Line 407) The introduction should be updated. Particularly, it should be the comparison between the predicted value and the actual value of the skew angle of different content images.

Round 2

Reviewer 1 Report

I appreciate the great efforts that the authors have made in response to my questions and concerns. I am satisfied with the refined manuscript. 

Reviewer 3 Report

1.       (Line 344)The authors dont mention the device used for the algorithm's time consumption. Please clarify this point.

2.       (Line 392) The authors should point out the clear rule of image classification, that is, how to judge the success or failure of image classification algorithm. In addition, it is clearly wrong that when one image is misjudged, the accuracy of the two types of images will be affected. It should be “precision” rather than “accuracy”. Please clarify the difference between “accuracy” and “precision”.